# A Combination of Rosa Multiflora and Zizyphus Jujuba Enhance Sleep Quality in Anesthesia-Induced Mice

**DOI:** 10.3390/ijms232214177

**Published:** 2022-11-16

**Authors:** Sanung Eom, Shinhui Lee, Jiwon Lee, Sung-Oh Sohn, Junho H. Lee, Jaeman Park

**Affiliations:** 1Department of Biotechnology, Chonnam National University, Gwangju 61186, Republic of Korea; 2R&D Center, Nature Pure Korea, Damyang 57309, Republic of Korea; 3Gilbut Korean Medicine Clinic, Seongnam 13144, Republic of Korea

**Keywords:** NCP—*Rosa multiflora Thunb. (Yeongsil)/Zizyphus jujuba Miller (Sanjoin)* natural combination product, sleep, Traditional Korean Medicine, melatonin receptor, *Rosa multiflora Thunb. (Yeongsil)*, SCN—suprachiasmatic nucleus, biological clock

## Abstract

Sleep is an essential component of quality of life. The majority of people experience sleep problems that impact their quality of life. Melatonin is currently a representative sleep aid. However, it is classified as a prescription drug in most countries, and consumers cannot purchase it to improve their sleep. This sleep induction experiment in mice aimed to identify a natural combination product (NCP) that can create synergistic sleep-promoting effects. Based on the mechanism of action of sleep, we investigated whether phenomenological indicators of sleep quality change according to the intake of NCP. The sleep onset and sleep time of the mice that consumed the NCP found by this study were improved compared to the existing sleep aids. The mean melatonin level in the blood increased by 197% compared to the control. To our knowledge, this is the first study to demonstrate that *Rosa multiflora Thunb. (Yeongsil)* can promote sleep similarly to *Zizyphus jujuba Miller (Sanjoin)*. The results indicate a preclinical study of NCPs containing *Rosa multiflora Thunb* and *Zizyphus jujuba Miller* developed by us showed significant differences in sleep incubation and duration depending on melatonin concentrations. Our results also suggest that increased melatonin concentrations in the blood are likely to improve sleep quality, especially regarding incubation periods.

## 1. Introduction

Since sleep disorders are a social problem that most people experience at some point in life, their effects are commonly felt. The incidences of headaches, chronic fatigue, and bipolar disorder have increased since the industrial revolution [1]. A lack of sleep lowers one’s quality of life. Accordingly, the number of patients with mental disorders such as depression, obsessive-compulsive disorder, and anxiety disorder is increasing annually; severe cases lead to problems such as suicide [2]. This leads to individual productivity losses and, ultimately, economic losses that are difficult to measure objectively [1,2,3].

Sleep is an essential part of life and plays an essential role in the cellular survival of all organs, such as recovering energy consumed during the day, healing cellular damage, and removing impurities [4,5,6]. Sleep is especially important for the brain, as it enables processing and editing functions to store new memories and delete old ones [7,8,9]. Sleep quality affects concentration, memory, emotional control, motivation, and stress management [10,11]. The biorhythm of sleep and wakefulness is based on a series of regulatory mechanisms that receive light information [12] through the optic nerve from the brain’s suprachiasmatic nucleus (SCN) and secrete dopamine and melatonin [13,14,15,16,17]. During daylight hours, the body’s dopamine level increases and melatonin level decreases, while at night, the dopamine level decreases and the melatonin level increases [13,18].

Studies to date have reported that, although γ-aminobutyric acid (GABA), serotonin, dopamine, and melatonin are involved in regulating the biological clock [19], melatonin is the most important hormone for sleep [11,20,21]. Since melatonin is widely used [22], it is commonly used in the United States. However, in many other countries, melatonin is still classified as a prescription drug, so consumers cannot purchase it directly. Therefore, in countries where melatonin cannot be purchased, there is a need for healthy, functional foods that naturally increase melatonin levels.

Most of the products currently used as sleep aids consist of Traditional Korean Medicines such as Zizyphus jujuba Miller (Sanjoin), Ulmus macrocarpa Hance (Yubaekpi), Rosa multiflora Thunb, (Yeongsil), nutmeg, gardenia, and Coptis chinesis Franch (Hwangryeon) that are used to treat sleep disorders in Traditional Korean Medicine [23]. However, scientific evidence of their sleep-promoting effects has not been studied. Of these products, only Zizyphus jujuba Miller (Sanjoin) has been found to help sleep through several studies [24,25]. A typical method for measuring a substance’s sleep-promoting efficacy involves observing changes in sleep latency and duration in mice using pentobarbital, tiletamine-zolazepam/xylazine, or other substances [26,27]. Therefore, here we first selected the 6 Traditional Korean Medicines candidate for NCP that induced significant changes in sleep latency and duration in the mouse. We then observed whether a synergistic effect appeared when Traditional Korean Medicines were consumed in different combinations and determined the melatonin and serotonin concentrations in the blood to identify the NCP with the highest sleep-inducing efficacy. Various sleep-improving natural substances have been discovered, and industrialization is underway. However, problems such as toxicity, side effects, and insufficient efficacy delay the development of new health-functional foods or medicines. This study verified that oriental medicine materials were used and already drastically increased the effect of sleep improvement, suggesting the development of efficient materials for sleep improvement.

Using this method, we collected novel scientific evidence showing that the natural product *Rosa multiflora Thunb (Yeongsil)* promotes sleep. Moreover, material analysis using high-performance liquid chromatography (HPLC) and liquid chromatography-mass spectrometry (LC-MS)identified active ingredients of the NCP. Furthermore, through a mouse behavioral experiment, we confirmed whether symptoms of anxiety were improved due to improved sleep quality. Therefore, this validation study aimed to measure the effects of NCP on sleep quality in mice through melatonin mechanisms.

## 2. Materials and Methods

### 2.1. Materials

*Rosa multiflora Thunb, (Yeongsil)* (Republic of Korea), *Zizyphus jujuba Miller (Sanjoin)* (Myanmar), *Ulmus macrocarpa Hance (Yubaekpi)* (Republic of Korea), *Hwangryeon* (Republic of Korea), gardenia (Republic of Korea), nutmeg (Republic of Korea), and *Sleep^n^* (Cosmax Bio Co., Gyeonggi-do, Republic of Korea) used in the experiment were purchased, confirmed as genuine through genetic differentiation, pulverized into powder, and passed through a standard 40 mesh sieve. *Rosa multiflora Thunb* and *Zizyphus jujuba Miller* were purchased from Chungmyuong Herb (#9188) and Midland Shwepyitrading co ltd (#148-1). During processing, filter cloths for extraction (HB, Daegu, Republic of Korea) and Advantec no. 1 filter paper (Toyo Roshi Kaisha Ltd., Tokyo, Japan) were used. During the analysis, Sigma-Aldrich (St. Louis, MO, USA), Chemface (Wuhan, Hubei, China), Deoksan (Ansan, Gyeonggido, Republic of Korea), and Daejung Chemical (Siheung, Gyeonggido, Republic of Korea) reagents were used. For the animal experiments, BALB/c mice, feed, and bedding (Orient Co., Seongnam, Gyeonggido, Republic of Korea) were used. Through this, the standardization result of the following extraction conditions was derived; The extraction temperature is 50 °C, the extraction solvent is 50% ethanol, the extraction time is 4 h, and the powder mixing ratio is the ratio of 6: 4 of *Rosa multiflora Thunb. (Yeongsil)*: *Zizyphus jujuba Miller (Sanjoin)*. The extract was filtered through Whatman filter paper, and the supernatants were evaporated in a rotary evaporator and lyophilized. Rosa multiflora used dried berries, and Zizyphus jujubas used dried seeds.

### 2.2. High-Performance Thin-Layer Liquid Chromatography and HPLC/liquid Chromatography-Mass Spectrometry (LC-MS)

High-performance thin-layer liquid chromatography (HPTLC) was performed to quickly and qualitatively analyze the indicator substance of *Zizyphus jujuba Miller*. For the extraction method, 1 g of powder was weighed, 20 mL of 50% methanol was added, and an ultrasonic extraction was run for 30 min, followed by filtration to obtain a standard solution of *Zizyphus jujuba Miller*. The adsorbent was HPTLC-Silica gel 60 F254, while the mobile phase solvent was saturated butanol. In addition, 5% vanillin-sulfuric acid was used as the color reagent, and the pattern was compared with the standard spinosin (C_28_H_32_O_15_, 608.5 g/mol) at detection wavelengths of 254 nm and 365 nm. HPTLC was performed to quickly and qualitatively analyze the indicator substance of *Rosa multiflora Thunb.* Since there is no standard analysis method for *Rosa multiflora Thunb*, the analysis method of *Zizyphus jujuba Miller* was applied. For the extraction method, 1 g of powder was weighed, 20 mL of 50% methanol was added, and an ultrasonic extraction was run for 30 min, followed by filtration to obtain a standard solution of *Rosa multiflora Thunb*. The adsorbent was HPTLC-Silica gel 60 F254, while the mobile phase solvent was saturated butanol. In addition, 5% vanillin-sulfuric acid was used as the color reagent, and the patterns were compared with the standard hyperoside (C_21_H_2O_O_12_, 464.4 g/mol) at detection wavelengths of 254 and 365 nm.

HPLC/liquid chromatography-mass spectrometry analysis was performed to analyze the active ingredients of *Rosa multiflora Thunb/Zizyphus jujuba Miller* natural combination product (NCP) and to verify the exact composition of *Rosa multiflora Thunb.* LC/MS consists of a liquid chromatograph, an ionizer, and a mass spectrometer. The mixture is separated on a liquid chromatograph, and the individual analyses separated are converted into ions in an ionizer. These ions are separated by a mass/charge ratio and appear as a spectrum in a mass spectrometer.

### 2.3. Animal Maintenance

For the animal experiments, Balb/c 10-week-old mice (male/female) were selected and stabilized for one week. Breeding conditions excluded factors that can affect sleep; that is, they included a 12-h light/12-h dark cycle, limited noise (40 dB or less), dim lighting (200–300 LUX 12 h/day), constant temperature (25 ± 2 °C), and mid-level humidity (55 ± 15%). The experiment included 8–10 animals per experimental group with 3–4 animals per cage with ample space, natural access, and sufficient feed and water provided until the start of the experiment. All experiments were conducted following the policy of the Laboratory Animal Ethics Committee and regulations related to animal testing (approval number: CNU IACUC-YB-2016-07).

### 2.4. Tiletamine-Zolazepam/Xylazine-Induced Sleep Experiment in Mouse

Sleep induction experiments were performed in triplicate using the same method and environment. Each group consisted of about 8–14 animals. All experiments were performed between 13:00 and 18:00. All mice were maintained on a 12 h light/12 h dark cycle (lights on at 9:00 am) and had free access to food and water during the sleep induction experiment. Mice ingested the test solution once a day by oral administration for one week. 30 min after the last oral administration, Tiletamine-zolazepam/Xylazine was intraperitoneally (I.P.) injected into mice. For tiletamine-zolazepam, Zoletil^®^ (Virbac Korea Co., Seoul, Republic of Korea) was used, and for Xylazine, Rompun^®^ (Bayer Korea Co., Gyeonggi-do, Republic of Korea) was used. Tiletamine-zolazepam was dissolved in PBS at 20 mg/kg and Xylazine at 5 mg/kg and injected intraperitoneally at 10 uL/g. Following injection, the mice were placed individual cages and monitored for measurements of sleep duration. For a better understanding of the sleep test in mice, the experimental procedures and timeline are shown in Figure 1E. Sleep latency time was recorded from the time of injection to the time of loss of the righting reflex, and sleep duration was defined as the difference in time between the loss and recovery of the righting reflex. The recovery time of the righting reflex was measured as the time all four feet touched the ground.

### 2.5. Enzyme-Linked Immunosorbent Assay (ELISA)

For the melatonin analysis, samples were orally administered to the mouse, and blood was collected by external jugular vein blood sampling 30 m later. The collected blood samples were coagulated at room temperature (20–25 °C) for 2 h or more and centrifuged at 1000× *g* for 30 min; the plasma was then separated, and 100 μL of cold ethyl acetate and 100 μL of the plasma sample were mixed. After vacuum centrifugation of the sample, the dried extract was resuspended in 100 μL of 1× kit stabilizer, and an analysis was performed according to the basic manual of the assay kit. A 100-μL aliquot of each standard, control, and plasma was added to the microplate, covered, mixed, and incubated at room temperature for 90 min. The plate was washed five times with 300 μL of washing solution, and 100 μL of 1× biotinylated antibody was added; the plate was then covered and incubated at room temperature for 1 h. The plate was washed five times with 300 μL of washing solution, 100 μL of 3,3′,5,5′-tetramethylbenzidine one-step substrate reagent was added, and the plate was incubated at room temperature in the dark for 30 min. The reaction was stopped by adding 100 μL of stop solution to the plate, and the change in absorbance at 450 nm was measured.

### 2.6. Cellular Experiment

To assess cell viability in vitro, cells treated with serially diluted samples were cultured for a certain period, and then 3-(4,5-dimethylthiazol-2-yl)-2,5-diphenyl-2H-tetrazolium bromide MTT, a reagent that reacts with the dehydrogenase of the mitochondria was used. Raw 264.7 cells were seeded in 96-well plates (4–6 × 10^4^ cells/well/100 μL), incubated for 24 h, and then incubated for 24 h by treatment with extracts at specific concentrations. After removal of the medium, 100 μL of fresh medium and 10 μL of a 12 mM MTT stock solution (negative control: 100 μL medium + 10 μL of MTT stock) was added, 25 μL of the medium was removed, 50 μL of dimethyl sulfoxide was added, the solution mixed, and the vial was incubated at 37 °C for 10 min. Absorbance was measured at 540 nm using a plate reader. The results are expressed as a percentage of cells that are 90% viable.

### 2.7. Open-Field Test

The basic activity level of each experimental animal was measured by measuring the search behavior that occurred upon exposure to a new space, determining the general level of walking activity, and measuring the change in motility, a characteristic stress-induced symptom. The stress level is assessed by measuring the starting latency, activity in the center, activity in the periphery, forefoot lifting, and grooming behavior. The higher the anxiety, the stronger an animal’s tendency to avoid open spaces. The open-field test (OFT) uses a box made of transparent acrylic placed under a white light source and tracks the movement of an experimental animal using an infrared detector. Such a program can analyze gait activity, repetitive activity, and standing by the number and time. The behavior pattern that appears when the experimental animal is first exposed to a large space is innate and distinct from the general learned behavior. It mainly compares the time the mice stay in the center with the time they wander at the edge. Taking advantage of the innate tendency to hover around the edges when a mouse is anxious, the time spent on the edge is a measure of anxiety. We measured the movement of the mouse on a square-shaped test apparatus of 20 cm in width and 20 cm in height for 6 min. The blue square in the center is a width of 10 cm and a vertical of 10 cm.

### 2.8. Statistical Analysis

Analysis of the screening test results can be determined through the ANOVA test. The ANOVA test is a method that is typically used to confirm the fit of the model. In particular, the *p*-value derived from this result is a major tool to confirm the importance of each coefficient. All data are represented as mean ± standard error of the mean (SEM). The significance test between the two groups was considered statistically significant when the *p*-value < 0.05. In this case, the *p*-value was calculated using the Student’s *t*-test. One-way ANOVA was used to test significance in three or more groups, and the Fisher-LSD method was used for post hoc testing.

## 3. Results

### 3.1. HPLC/Liquid Chromatography–Mass Spectrometry Analysis and Tiletamine-Zolazepam/Xylazine Sleep Induction Model Mouse for Zizyphus Jujuba Miller (Sanjoin)/Rosa Multiflora Thunb (Yeongsil) Natural Combination Product

HPLC and liquid chromatography-mass spectrometry (LC-MS) were used to analyze the active ingredients of *Rosa multiflora Thunb (Yeongsil)/Zizyphus jujuba Miller (Sanjoin)* natural combination product (NCP). LC-MS was performed to verify the exact composition of *Rosa multiflora Thunb (Yeongsil)*, which included the following four main substances: RT 6.55 min, 465.10 *m*/*z*, hyperoside; RT 6.65 min, 479.08 *m*/*z*, Miquelianin; RT 7.86 min, 611.16 *m*/*z*, multinoside A; and RT 10.23 min, 653.17 *m*/*z*, multinoside A acetate (Figure 1A). LC-MS was performed to verify the exact composition of *Zizyphus jujuba Miller (Sanjoin)*, which included the following two main substances: RT 6.71 min, 609.18 *m*/*z*, spinosin; RT 9.17 min, 785.23 *m*/*z*, spinosin B (Figure 1B). LC-MS was performed to verify the exact composition of *Rosa multiflora Thunb (Yeongsil)/Zizyphus jujuba Miller (Sanjoin)* NCP, which included the following five main substances: RT 6.61 min, 479.08 *m*/*z*, Miquelianin; RT 6.69 min, 609.18 *m*/*z*, spinosin; RT 7.83 min, 611.16 *m*/*z*, multinoside A; RT 9.14 min, 785.23 *m*/*z*, spinosin B; RT 10.22 min, 653.17 *m*/*z*, multinoside A acetate (Figure 1C). In HPLC, the main constituents—spinosin, multinoside A, hyperoside, multinoside A acetate, and Miquelianin (quercitrin 3-O-glucuronide) –were confirmed as stable during processing (Figure 1D). Next, we conducted a sleep induction experiment using a Tiletamine-zolazepam/Xylazine sleep induction model mouse to examine whether the intake of NCP affected the phenomenal symptoms of sleep initiation and duration. We developed NCP with synergistic effects through sleep behavior experiments in three steps. All sleep induction experiments used the same method and handling as the pipeline above (Figure 1E).

### 3.2. Selection of Natural Product Candidates for Improving Sleep and Biological Changes in Mice Following Oral Administration of Traditional Korean Medicine

We conducted the first sleep induction experiment to identify natural product candidates effective in the pre-clinic stage. Natural product candidates were selected from those known to be effective for sleep in Traditional Korean Medicine. Moreover, the weight of the mice before and after oral administration at 50 mg/kg of each Traditional Korean Medicine was observed (Figure 2A,B). As a result, the change in body weight after administration did not significantly change compared to before. In addition, no specific abnormalities were found in the amount of excretion, drinking water, food intake, and gross pathological examination. These results suggest that oral administration of 50 mg/kg of Traditional Korean Medicine in mice does not show acute toxicity. As a positive control, we used *Zizyphus jujuba Miller (Sanjoin)*, a natural product shown to help sleep in previous studies, and *Sleep^n^*, an existing sleep aid product certified by the Korean Ministry of Food and Drug Safety as a health functional food that can help sleep (Figure 2C,D). The average sleep latency of the mouse that 2.66 ± 0.29 min for saline and 2.11 ± 0.17 min for *Hwangryun*, 1.36 ± 0.19 min for *Rosa multiflora Thunb. (Yeongsil)*, 1.96 ± 0.31 min for gardenia, 1.98 ± 0.25 min for nutmeg, 1.31 ± 0.27 min for *Ulmus macrocarpa Hance (Yubaekpi)*, 1.33 ± 0.18 min for *Zizyphus jujuba Miller (Sanjoin)*, and 1.74 ± 0.21 min for *Sleep^n^*. The efficacy of *Rosa multiflora Thunb. (Yeongsil)* appeared similar to that of *Sleep^n^* and *Zizyphus jujuba Miller (Sanjoin)*. The sleep duration was measured as a mean of 53.5 ± 5.89 min for the control group, 61.2 ± 4.99 min for *Coptis chinesis Franch (Hwangryeon)*, 81.5 ± 7.53 min for *Rosa multiflora Thunb. (Yeongsil)*, 57.7 ± 8.68 min for gardenia, 64.2 ± 6.12 min for nutmeg, 65.9 ± 8.10 min for *Ulmus macrocarpa Hance (Yubaekpi)*, 80.7 ± 9.21 min for *Zizyphus jujuba Miller (Sanjoin)* (*p*-value < 0.001), and 72.1 ± 5.56 min for *Sleep^n^* (*p*-value = 0.012). The sleep duration showed a similar pattern as sleep latency. These results are meaningful in that they are the first to suggest that *Rosa multiflora Thunb. (Yeongsil)* can help sleep (*p*-value < 0.001).

### 3.3. Mouse Sleep Experiment by Concentration for Each Combination of Traditional Korean Medicine

To develop a combination that can exert a synergistic effect, a second sleep induction experiment (Figure 3A,B) was conducted with different combinations of *Rosa multiflora Thunb. (Yeongsil)*, *Zizyphus jujuba Miller (Sanjoin)* and *Ulmus macrocarpa Hance (Yubaekpi)* were selected in the first sleep induction experiment. A negative control was used as an excipient (maltodextrin) for commercialization. As a result, the average sleep latency with the combination of *Rosa multiflora Thunb. (Yeongsil)* and *Zizyphus jujuba Miller (Sanjoin)* was 0.83 ± 0.12 min, while that of *Zizyphus jujuba Miller (Sanjoin)* only (PC) was a mean 1.85 ± 0.16 min. The average sleep duration with the combination of *Rosa multiflora Thunb. (Yeongsil)* and *Zizyphus jujuba Miller (Sanjoin)* was 96.98 ± 7.32 min. The average sleep duration of PC was 54.20 ± 6.00 min. The loss time of lighting reflex was shortened by 223%, while the time to recover of lighting reflex increased by 179% compared to the mice fed with *Zizyphus jujuba Miller (Sanjoin)* only. These results indicate that the sleep latency in the mouse more than doubled, while the sleep duration approximately doubled. Thus, we discovered the combination of *Rosa multiflora Thunb. (Yeongsil)* and *Zizyphus jujuba Miller (Sanjoin)* produced a synergistic effect and conducted the third sleep induction experiment (Figure 3C,D) to verify the effect according to the concentration of this NCP. The experimental method was performed in the same manner. The mouse fed with 0.3 mg/kg NCP showed an average sleep latency of 1.74 ± 0.10 min and an average sleep duration of 72.96 ± 8.32 min, those fed with the 1 mg/kg NCP had an average sleep latency of 1.37 ± 0.15 min and an average sleep duration of 102.23 ± 7.34 min, those fed with the 3 mg/kg NCP had an average sleep latency of 1.52 ± 0.23 min and average sleep duration of 82.73 ± 4.62 min, and those fed with the 10 mg/kg NCP had an average sleep latency of 1.20 ± 0.22 min and average sleep duration of 88.10 ± 11.44 min. Thus, the NCP showed concentration-dependent effects. The most efficient concentration showing a significant difference was 1 mg/kg (*p*-value < 0.001). Significance was higher than that of the group administered orally at 50 mg/kg of *Sleep^n^* (*p*-value = 0.002).

### 3.4. Blood Melatonin and Serotonin Concentrations and Cytotoxicity of Mice That Ingested with Optimal NCP Ingredients

We performed melatonin and serotonin ELISA to verify whether the cause of the shortened elevation time and increased sleep duration in mice fed with the NCP was based on the melatonin logic mechanism. We collected serum samples through external competitive vein blood sampling from mice fed with the NCP once a day for five days. On the last day, blood samples were collected 30 min after oral administration. The ELISA test was conducted immediately on the same day the blood was drawn. The average melatonin level of the control group was 773.95 ± 89.38 pg/mL; the *Sleep^n^* group, 1418.57 ± 192.40 pg/mL; the 0.3 mg/kg NCP group, 1059.44 ± 97.21 pg/mL; the 1 mg/kg NCP group, 1526.40 ± 285.80 pg/mL; and the 10 mg/kg NCP group, 1518.49 ± 288.38 pg/mL (Figure 4A). The levels of the positive control and 1 mg/kg groups have no difference significantly. These results suggest that the consumption of this NCP may effectively increase the melatonin concentration in the blood. Furthermore, we performed serotonin ELISA to determine whether serotonin, which is used as a precursor of melatonin, also increases when NCP is ingested. Serum serotonin levels in the control group were 273.71 ± 51.36 ng/mL; the *Sleep^n^* group, 520.62 ± 41.04 ng/mL; the 0.3 mg/kg NCP group, 451.91 ± 35.01 ng/mL; the 1 mg/kg NCP group, 476.49 ± 23.47 ng/mL; the 10 mg/kg NCP group, 574.21 ± 68.58 ng/mL (Figure 4B). There was a significant difference with a *p*-value of 0.003 between the control group and the 10 mg/kg group through one-way ANOVA tested post hoc by the Fisher-LSD method. These results mean that oral NCP intake can increase the blood’s serotonin concentration. Above all, it should be noted that the increased pattern of blood serotonin is similar to the increased pattern of blood melatonin concentration. A significant difference in the toxicity test using raw 264.7 cells was observed with a *p*-value of 0.001 from 0.4g/kg (Figure 4C). The highest treatment concentration that secured safety was 0.39 mg/g with IC_90_.

### 3.5. Open-Field Test According to NCP Intake

We confirmed that the NCP developed through this study helps sleep better. Furthermore, OFT was performed to check whether it can help relieve anxiety by improving sleep quality. OFT was performed through the apparatus and method described in the section on the method. The 50 mg/kg *Sleep^n^* was used as a positive control group, and one-way ANOVA compared significant differences with the control group fed the same volume of saline. Figure 5A,B was the result of measuring the time they stayed in the place periphery and center of the mouse in a separate square cage for 6 min as a percentage, and Figure 5C tracks the overall movement pattern. In NCP-fed mouse groups, the time at the periphery was decreased, and the time at the center increased with increasing concentration, as shown in Figure 5A,B. In Figure 5C, In the blue square shape, which is the central part of the cage, there were more movement lines in the NCP group than in the control and the *Sleep^n^* group. Figure 5A, as a result of measuring the percentage of time that mouse stayed at the periphery of an independent square cage, control group, 88.00 ± 2.46%; *Sleep^n^* group, 81.73 ± 3.90%; 0.3 mg/kg NCP group, 73.43 ± 4.69%; 1 mg/kg NCP group, 71.65 ± 3.62%; 5 mg/kg NCP group, 70.76 ± 3.41; and 10 mg/kg NCP group, 66.96 ± 7.13%, it confirmed that the NCP fed group less stayed in the periphery compared to the control and *Sleep^n^*. Figure 5B, as a result of measuring the percentage of time the mouse stayed in the center of an independent square cage, the control group, 12.00 ± 2.46%; *Sleep^n^* group, 17.97 ± 2.30%; 0.3 mg/kg NCP group, 26.57 ± 4.69%; 1 mg/kg NCP group, 28.35 ± 3.62%; 5 mg/kg NCP group, 29.24 ± 3.41%; And 10 mg/kg NCP group, 33.04 ± 7.13%, compared to the control and *Sleep^n^* group, it confirmed that the NCP fed mouse group more stayed in the center. Figure 5D,E was the result of measuring the mouse activity in an independent square cage for 6 min. Figure 5F, results from measuring the mouse activity in the cage. The central movement is indicated in red, and the peripheral movement is indicated in yellow. Figure 5D showed a percentage by measuring the mouse activity in the periphery zone, the result of the control group, 78.71 ± 1.97%; sleep group, 78.17 ± 2.57%; 0.3 mg/kg NCP group. 75.30 ± 1.66%; 1 mg/kg NCP group, 75.13 ± 1.94%; 5 mg/kg NCP group, 71.96 ± 2.99%; And 10 mg/kg NCP group, 74.06 ± 3.58%, it was confirmed that there was less movement in the periphery in the NCP fed mouse groups compared to the control and *Sleep^n^*. Figure 5E showed a percentage by measuring mouse activity in the center zone, the control group, 20.14 ± 1.89%; sleep group, 21.84 ± 2.57%; 0.3 mg/kg NCP group, 24.70 ± 1.66%; 1 mg/kg NCP group, 24.87 ± 1.94%; 5 mg/kg NCP group, 28.04 ± 2.99%; and 10 mg/kg NCP group, 25.94 ± 3.58%, it was confirmed that the more activity in the center was NCP fed mouse group compared to the control and *Sleep^n^*. The intensity of depression depends on the time they stay at which site in the cage and how long, and it is expressed as a lower depression as it stays in the center. Therefore, it was confirmed that depression was reduced in all NCP-fed mouse groups.

## 4. Discussion

After the invention of electricity, people began to complain of sleep problems that disrupted their biological clocks by being able to see light at night. In modern times, as the distinction between day and night has become more blurred, about 20–30% of adults have chronic insomnia, and most adults experience pain from sleep problems [28,29]. However, there are no available treatments that do not require a prescription and no effective health-functional foods (sleep supplements) that improve sleep quality and common sleep problems. Therefore, in this study, we developed an NCPeffective for sleep in the pre-clinic stage with a gradual approach. We demonstrated that the oral administration of 0.3 mg/kg of the NCP (*Rosa multiflora Thunb. (Yeongsil)*:*Zizyphus jujuba Miller (Sanjoin),* 6:4) to a Tiletamine-zolazepam/Xylazine sleep-induced mouse model improved sleep quality. In particular, the experimental results in the in vivo stage showed for the first time that *Rosa multiflora Thunb. (Yeongsil)* can help sleep, and when *Zizyphus jujuba Miller (Sanjoin)* and *Rosa multiflora Thunb. (Yeongsil)* were consumed alone, and a mixture thereof was consumed in a 6:4 ratio; indices of more than 223% were obtained. The aforementioned results prove that the oral intake of NCP effectively improves sleep.

According to the research results to date, many studies have reported that *Zizyphus jujuba Miller (Sanjoin)*—Ziziphus spinosa seeds (*Ziziphus jujuba Mill*. var. spinosa [Bunge] Hu ex HF Chou; hereafter *Zizyphus jujuba Miller (Sanjoin)*) in Traditional Korean Medicine is effective for insomnia [25,30]. However, the representative ingredients of *Zizyphus jujuba Miller (Sanjoin)* cannot cross the blood-brain barrier (BBB) [30]. Since these active ingredients of *Zizyphus jujuba Miller (Sanjoin)* did not pass the BBB, its mechanism of action was unclear. A recent study reported that *Zizyphus jujuba Miller (Sanjoin)* does not act directly on the central nervous system; rather, it improves insomnia by restoring the balance of intestinal microbes and controlling their metabolism [30]. Another study suggested that the ingredients interact with GABA receptors and may be involved in sleep through the GABA mechanism [31]. Another study reported that it is effective for sleep in the clinical trial stage [32]. However, there are no studies on the relationship between *Rosa multiflora Thunb. (Yeongsil)* and sleep. The present paper is the first study to present evidence that *Rosa multiflora Thunb. (Yeongsil)* can help improve sleep. We found similar degrees of sleep improvement in mice fed with *Rosa multiflora Thunb. (Yeongsil)* under the same conditions as that of mice fed *Zizyphus jujuba Miller (Sanjoin)* once a day for seven days (Figure 2). It is clear that research on *Rosa multiflora Thunb. (Yeongsil)* is required to increase our understanding of its mechanism.

As previously mentioned, *Zizyphus jujuba Miller (Sanjoin)*, which was shown to improve sleep through previous studies, and *Sleep^n^* (KR, Gyeonggi-do, Cosmax Bio), which has been certified as a health functional food and sleep aid by the Korean Ministry of Food and Drug Safety, were used as positive controls in this study. The ingredients of *Sleep^n^* are rice bran alcohol extract, vitamin B6, and niacin. Two positive controls were used to develop the NCP that is more effective than previous products and studies. As a result, it was more than 223% more effective when *Rosa multiflora Thunb. (Yeongsil)* and *Zizyphus jujuba Miller (Sanjoin)* were ingested together at a ratio of 6:4 than when either was consumed alone. The sleep latency was reduced by 179%, while the melatonin concentration was 107% higher than that of the *Sleep^n^* product (50 mg/kg of *Sleep^n^* vs. 1 mg/kg of NCP). These results indicate that several index components do not interact by a single mechanism but operate by at least two mechanisms related to sleep; these mechanisms can influence each other. In addition, sleep quality can be improved compared to that with existing products due to this synergistic effect. In past studies, sleep improvement effects using Nelumbo nucifera and Ecklonia cavatable brown seaweed have been studied.

We identified the main substances of *Rosa multiflora Thunb. (Yeongsil)*, *Zizyphus jujuba Miller (Sanjoin)*, and the NCP through LC-MS (Figure 1). There are four main substances in *Rosa multiflora Thunb. (Yeongsil)*: RT 6.55 min, 465.10 *m*/*z*, hyperoside; RT 6.65 min, 479.08 *m*/*z*, Miquelianin; RT 7.86 min, 611.16 *m*/*z*, multinoside A; and RT 10.23 min, 653.17 *m*/*z*, multinoside A acetate. Hyperoside protects BBB [33,34], and multinoside A and multinoside A acetate require biological research. In particular, Miquelianin was previously reported as being able to pass the BBB [35,36]. This result suggests the possibility that Miquelianin is a major component of *Rosa multiflora Thunb. (Yeongsil)*, it is delivered to the brain and is directly involved in the melatonin receptor (MT1) in the SNC [37], a part of the brain that receives light energy from the optic nerve and controls the biological clock [11]. During the daytime, the SNC suppresses the production of melatonin in the pineal gland; at night, it secretes melatonin into the bloodstream. MT1 and secreted melatonin, a full agonist, combine to deliver an electrical signal that signals the start of sleep throughout the nervous system. These electrical signals are most likely the key signal for entering the first non-rapid eye movement phase of sleep [38,39]. According to the results of this study, *Rosa multiflora Thunb. (Yeongsil)* is transmitted to the brain and is highly likely to interact with MT1, a neuroreceptor of the central clock. The fact that *Rosa multiflora Thunb. (Yeongsil)* and *Zizyphus jujuba Miller (Sanjoin)* have different mechanisms according to previous studies is consistent with the results of mice consuming the NCP of *Rosa multiflora Thunb. (Yeongsil)* and *Zizyphus jujuba Miller (Sanjoin)* show a synergistic effect on sleep improvement.

Although the mechanism of falling asleep is not yet precisely known, according to previous studies, the secretion of dopamine during the day and melatonin at night is a representative phenomenon of the circadian rhythm [18]. The main function of the SCN, the organ responsible for this circadian rhythm, is the central clock. It was long believed that the circadian rhythm was determined only by the central clock. However, intracellular molecular mechanisms through transcriptional regulation of clock/mbal1 protein in cells other than SCN were recently revealed [40,41], and the concept of a peripheral clock [42] was introduced. This means that there is a peripheral clock that can independently activate clock genes in other regions. Therefore, the concept that circadian rhythm is constituted by a hierarchical structure between the central and peripheral clocks at the individual mammalian level is widely accepted. Recently, as the neurobiological regulation mechanism in which the biological clock is regulated by MT1 activity expressed in SCN neurons was revealed, the development of sleep therapeutics through agonism of the MT1 receptor is attracting attention in the field of drug development through electro-neurophysiology. Melatonin has become a biomarker for sleep because it is the full agonist of the MT1 [20,43]. Others include GABA and cortisol. Melatonin has a half-life of 20–50 min and few side effects, so it is widely sold in the United States. Melatonin is a representative component of sleep. However, in all but a few countries, including the United States, melatonin is classified as a prescription drug and cannot be purchased by the general public. Thus, here we developed a sleep aid that improves sleep quality by increasing the concentration of melatonin in the blood.

Of course, this study also has limitations. First, it is unclear whether melatonin increases sleep duration based on melatonin mechanisms. In fact, the US Food and Drug Administration has not approved melatonin as a drug because it has a short half-life, and it is unclear whether it helps patients with chronic insomnia maintain sleep. Since the binding of melatonin and MT1 is a way to signal the onset of sleep, the US Food and Drug Administration proved its efficacy for sleep initiation. Second, since we did not directly experiment with a single substance in this study, we cannot be sure of the specific mechanism of action of *Rosa multiflora Thunb. (Yeongsil)*. For a similar reason, molecular-level data are lacking at the basic research stage. However, as the causal mechanisms of sleep have not yet been elucidated, phenomenological indicators were measured through mouse sleep experiments, and the efficacy was investigated based on the most plausible melatonin–MT1 activation mechanism. Due to these clear limitations, we analyzed the representative components; among them, we are planning a follow-up study to identify pharmacological indicators of the interaction with MT1 of the SCN and active ingredients of *Rosa multiflora Thunb. (Yeongsil)* through the electrophysiology technique.

## 5. Conclusions

According to this preclinic level study of the NCP containing *Rosa multiflora Thunb. (Yeongsil)* and *Zizyphus jujuba Miller (Sanjoin)* developed by us, significant differences in sleep latency and duration based on melatonin concentration were noted. This result indicates that *Rosa multiflora Thunb. (Yeongsil)* and *Zizyphus jujuba Miller (Sanjoin)* could act as a sleep aid for modern people suffering from sleep problems in countries where melatonin cannot be purchased. Our results also suggest that it is highly likely that increased concentrations of melatonin in the blood will improve sleep quality, especially in terms of latency. For 20–30% of adults with chronic insomnia, the use of a prescription drug that is effective for inducing and maintaining sleep is of interest. Due to this potential treatment for insomnia, follow-up studies on the representative ingredients of this NCP are required in various fields. In a follow-up study, we plan to identify candidate drugs by analyzing the receptor–ligand interaction mechanism using the two-electrode voltage-clamp technique as a neurophysiological approach to MT1 in the brain. Our findings suggest the need to prove a new molecular regulatory mechanism based on the representative components of this NCP.

## Figures and Tables

**Figure 1 ijms-23-14177-f001:**
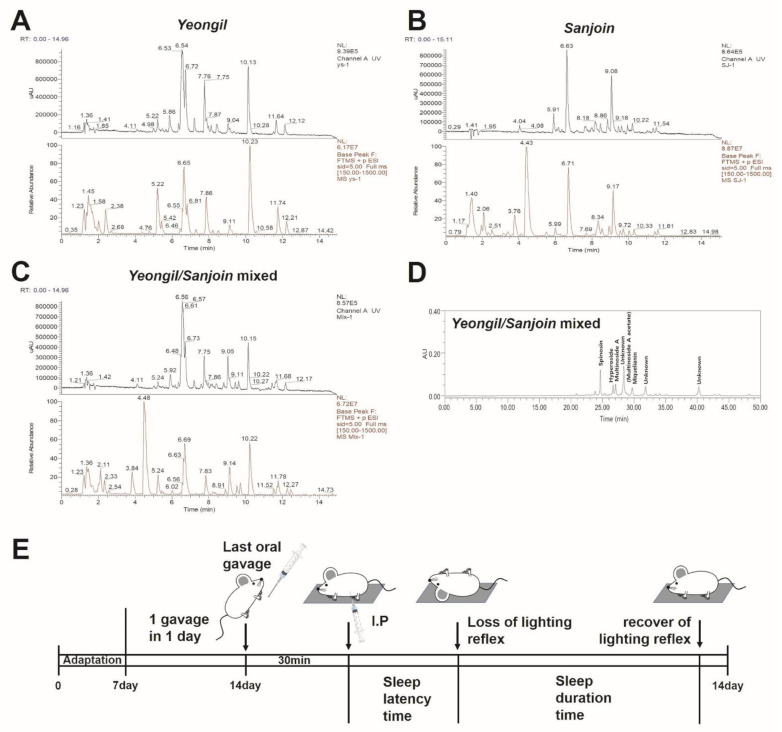
Confirmation of active ingredients through LC-MS analysis and anesthetic sleep. The LC-MS result of *Rosa multiflora Thunb. (Yeongsil)* (**A**). The LC-MS result of *Zizyphus jujuba Miller (Sanjoin)* (**B**). The LC-MS result of NCP mixed with *Rosa multiflora Thunb. (Yeongsil)* and *Zizyphus jujuba Miller (Sanjoin)* in a ratio of 6:4 (**C**). Quantitative results of NCP by HPLC (**D**). A schematic diagram of the pipeline of the sleep induction experiment (**E**).

**Figure 2 ijms-23-14177-f002:**
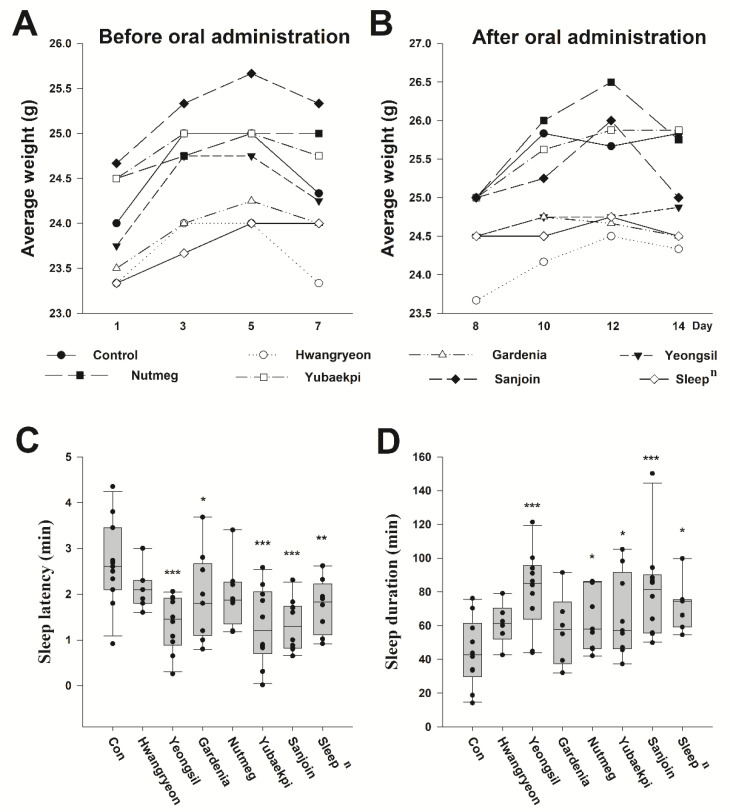
Changes in mice according to natural product intake. Changes in body weight of mice before taking Traditional Korean Medicine (**A**) and after starting oral administration (**B**). The sleep latency of mice fed once a day for seven days at 50 mg/kg (**C**) and the sleep duration result (**D**). For (**A**,**B**), the average value of each group was used. Significance was verified through one-way ANOVA, and significance was obtained between groups by post hoc testing using the Fisher-LSD method. The control group was administered saline in the same volume. *** *p* < 0.001, ** *p* < 0.01, * *p* < 0.05.

**Figure 3 ijms-23-14177-f003:**
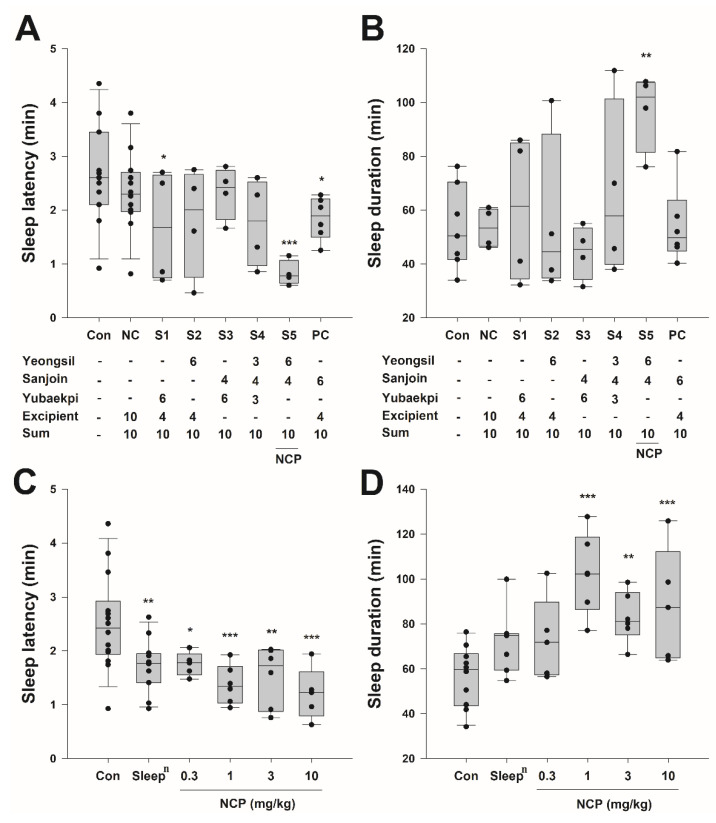
Optimal natural product combination screening. Changes in sleep latency (**A**) and sleep duration (**B**) in mice when different combinations were ingested. Sleep latency (**C**) and sleep duration (**D**) of mice fed NCP by concentration. All data were verified for significance through one-way ANOVA and post-test by Fisher-LSD method to obtain significance between groups. The control group is saline of the same volume. *** *p* < 0.001, ** *p* < 0.01, * *p* < 0.05.

**Figure 4 ijms-23-14177-f004:**
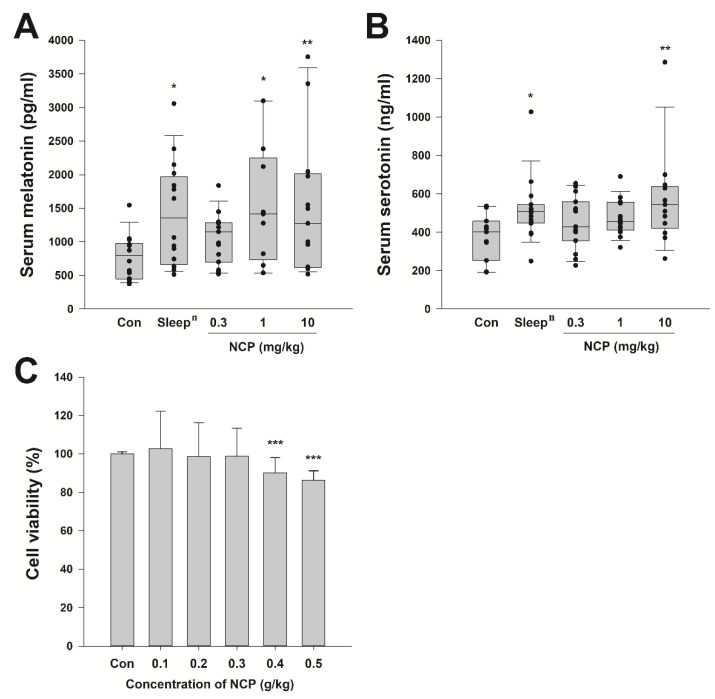
Sleep factor blood test in mice according to NCP intake and toxic concentration in cells. Serum melatonin levels (**A**) and serum serotonin levels (**B**) in mice when NCP was ingested by concentration. A toxicity test result of NCP using Raw 264.7 (**C**). All data were verified for significance through one-way ANOVA and post-test by Fisher-LSD method to obtain significance between groups. *** *p* < 0.001, ** *p* < 0.01, * *p* < 0.05.

**Figure 5 ijms-23-14177-f005:**
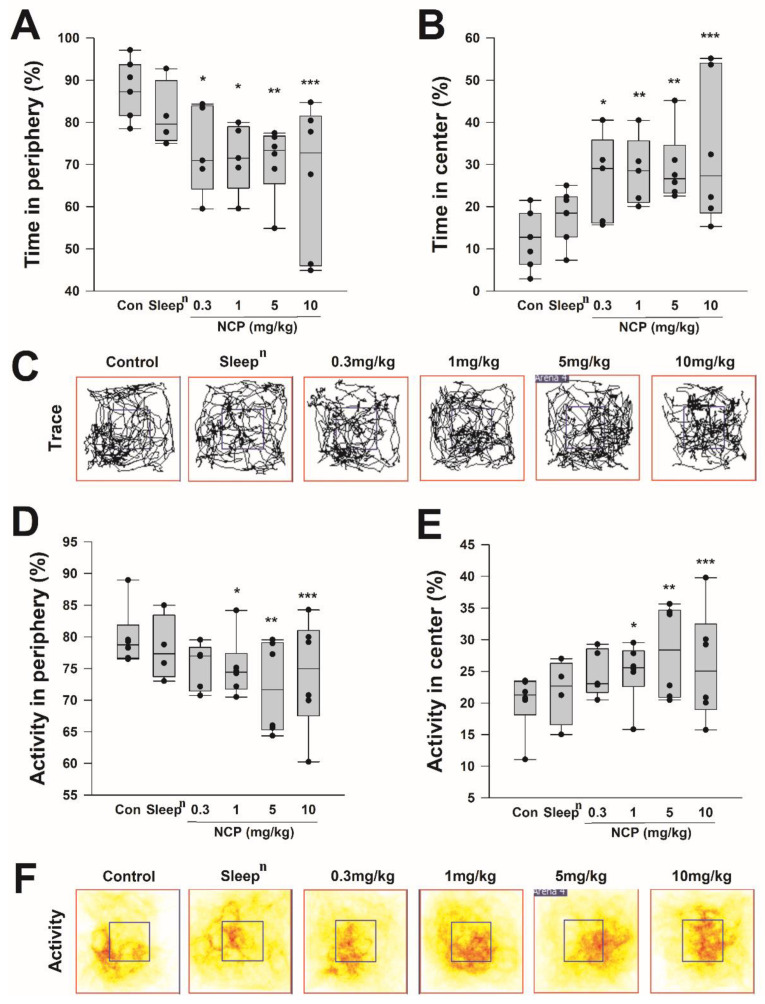
Anxiety and depression level test according to NCP intake in mice. OFT results of mice fed NCP by concentration; Time in periphery percentage (**A**) and time in center percentage (**B**). The entire trace of mice for 6 min (**C**). Activity at the edge (**D**) and activity at the center square (**E**). As a result of displaying the degree of activity in color (**F**), the more red the position, the more movement. Significance was verified through one-way ANOVA, and significance was obtained between groups by post hoc testing using the Fisher-LSD method. Only D and E were calculated with the Student’s *t*-test. The control group is saline of the same volume. *** *p* < 0.001, ** *p* < 0.01, * *p* < 0.05.

## Data Availability

Data is contained within the article.

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
