# Peer review of "A Combination of Rosa Multiflora and Zizyphus Jujuba Enhance Sleep Quality in Anesthesia-Induced Mice"

_ijms, 2022, doi:10.3390/ijms232214177_

Round 1

Reviewer 1 Report

The article entitled "A combination of Rosa multiflora and Zizyphus jujuba en-2 hance sleep quality in anesthesia-induced mice" is an interesting study on the effect of natural products to modulate sleep in animal models.

It is certainly the first study to demonstrate that Rosa multiflora Thunb. (Yeongsil) can promote sleep similarly to Zizyphus jujuba Miller (Sanjoin). The results indicate that the NCP comprising Rosa multiflora Thunb. (Yeongsil) and Zizyphus jujuba Miller (Sanjoin) has a synergistic effect, and it effectively improves sleep quality by increasing the concentration of melatonin in the blood. Therefore, the findings of this study suggest the need for basic and subsequent research on representative single-chemicals of NCP.

Overall, I think it's a very good article and very well written. I congratulate the authors for their scientific and methodological rigor. The use of graphs and figures is very useful for a better understanding of the data. The bibliography is also updated and consistent with the topic. In my opinion, there are no defects and the article must be accepted as submitted.

Author Response

First of all, thank you for seeing it positively. We revised the manuscript in a better way, thanks to your kindness which comments on the details. We hope this revised manuscript will meet your expectations. All revised parts highlighted in red. The manuscript is rewritten to make it easier for the reader to interpret and requested the recorrection from professional English editing company

Reviewer 2 Report

I applaud the authors for this work. A combination of Rosa multiflora and Zizyphus jujuba enhance sleep quality in anesthesia-induced mice.

Please make sure that the structure for citing published literature in the text, as well as the style of references in the References section, are consistent with the journal's style (see Instructions to Authors).

·         English language needs revision for style and syntax.

·         Abstract must be rewritten. I suggest focusing the abstract on your study and your results.

·         Why you chose 14 days?.

·         The experimental protocol must be clearer.

·         Please add the originality of the study and add hypothesis at the end of the introduction section. Be please be more specific.

·         Did authors perform other statistical analysis? Please be more specific. I suggest bonferroni test instead Fisher.

·         Please discuss the results of the study in relation to the previous studies.

Author Response

I applaud the authors for this work. A combination of Rosa multiflora and Zizyphus jujuba enhance sleep quality in anesthesia-induced mice.

Please make sure that the structure for citing published literature in the text, as well as the style of references in the References section, are consistent with the journal's style (see Instructions to Authors).

-English language needs revision for style and syntax.

Author response: First of all, thank you for seeing it positively. We revised the manuscript in a better way, thanks to your kindness which comments on the details. We hope this revised manuscript will meet your expectations. All revised parts highlighted in red. The manuscript is rewritten to make it easier for the reader to interpret and requested the recorrection from professional English editing company

Abstract must be rewritten. I suggest focusing the abstract on your study and your results.

Author response: The abstract is rewritten to make it focusing to our results at line 16-25.

Why you chose 14 days?

Author response: We referred to existing review and research journal papers to design the experimental method. It was designed with reference to information such as food intake and drinking water of mice. ( Behav Genet. 2002 November ; 32(6): 435–443, Nat Commun. 2017 Jul 24;8(1):155.)

The experimental protocol must be clearer.

Author response: We added furthermore detail protocol following advice at line 95-100 and 104-120.

Please add the originality of the study and add hypothesis at the end of the introduction section. Be please be more specific.

Author response: We added some sentences following advice at line 70-75.

Did authors perform other statistical analysis? Please be more specific. I suggest bonferroni test instead Fisher.

Author response: We used the ANOVA test for analysis of the results of the screening test.

Please discuss the results of the study in relation to the previous studies.

Author response: We added some sentences following advice at line 420-434.

Reviewer 3 Report

Acceptable. Hoever, limitation of the study should be indicated as detailes.

Author Response

Acceptable. Hoever, limitation of the study should be indicated as detailes.

Author response: First of all, thank you for seeing it positively. We revised the manuscript in a better way, thanks to your kindness which comments on the details. We hope this revised manuscript will meet your expectations. All revised parts highlighted in red. The limitation of the study is rewritten in introduction part.

Reviewer 4 Report

In the title write names of the plant species as per Binomial Nomenclature. Add some pictures of the Lab where studies was conducted. Its not clear how the plant species were  indentified  and where voucher specimens deposited for future reference? Clear this and if not give voucher numbers to plant species and submit in Herbarium 

Author Response

In the title write names of the plant species as per Binomial Nomenclature. Add some pictures of the Lab where studies was conducted. Its not clear how the plant species were indentified and where voucher specimens deposited for future reference? Clear this and if not give voucher numbers to plant species and submit in Herbarium

Author response: First of all, thank you for seeing it positively. We revised the manuscript in a better way, thanks to your kindness which comments on the details. We hope this revised manuscript will meet your expectations. All revised parts highlighted in red. The plant species is rewritten in Materials and Methods part at line 86-92.